# Improved Complementary Pulmonary Nodule Segmentation Model Based on Multi-Feature Fusion

**DOI:** 10.3390/e24121755

**Published:** 2022-11-30

**Authors:** Tiequn Tang, Feng Li, Minshan Jiang, Xunpeng Xia, Rongfu Zhang, Kailin Lin

**Affiliations:** 1School of Optical-Electrical and Computer Engineering, University of Shanghai for Science and Technology, Shanghai 200093, China; 2Key Laboratory of Optical Technology and Instrument for Medicine, Ministry of Education, University of Shanghai for Science and Technology, Shanghai 200093, China; 3School of Physics and Electronic Engineering, Fuyang Normal University, Fuyang 236037, China; 4Department of Biomedical Engineering, Florida International University, Miami, FL 33174, USA; 5Fudan University Shanghai Cancer Center, Shanghai 200032, China

**Keywords:** lung nodule segmentation, deep learning, cross-scale, complementary

## Abstract

Accurate segmentation of lung nodules from pulmonary computed tomography (CT) slices plays a vital role in the analysis and diagnosis of lung cancer. Convolutional Neural Networks (CNNs) have achieved state-of-the-art performance in the automatic segmentation of lung nodules. However, they are still challenged by the large diversity of segmentation targets, and the small inter-class variances between the nodule and its surrounding tissues. To tackle this issue, we propose a features complementary network according to the process of clinical diagnosis, which made full use of the complementarity and facilitation among lung nodule location information, global coarse area, and edge information. Specifically, we first consider the importance of global features of nodules in segmentation and propose a cross-scale weighted high-level feature decoder module. Then, we develop a low-level feature decoder module for edge feature refinement. Finally, we construct a complementary module to make information complement and promote each other. Furthermore, we weight pixels located at the nodule edge on the loss function and add an edge supervision to the deep supervision, both of which emphasize the importance of edges in segmentation. The experimental results demonstrate that our model achieves robust pulmonary nodule segmentation and more accurate edge segmentation.

## 1. Introduction

Lung cancer remains the most fatal type of cancer worldwide [1], and among Chinese women (most non-smokers), the incidence and mortality of lung cancer rank first in the world [2]. Early screening of lung cancer is key to improving the survival chances of patients. In clinical practice, the most widely used predictors for assessing the probability of malignancy and tumor progression are the size, shape, and growth rate of the nodule [3]; therefore, accurate segmentation of lung nodules is essential in the diagnosis of lung cancer. However, with the popularization of pulmonary computed tomography (CT), the feasibility of manual CT annotation has become increasingly unrealistic because it is too strenuous [4]. Therefore, it is necessary to develop a computer-aided diagnosis (CAD) of lung nodule segmentation to avoid laborious manual annotation in clinical practice, which objectively ensures the consistency of nodule diagnosis [5].

As shown in Figure 1, lung nodules have high variability in size, shape, and intensity, as well as similar visual characteristics between nodules and their surroundings. Although the traditional semi-automatic segmentation methods based on image processing improve the repeatability of annotation, they generally suffer from poor adaptability and low segmentation accuracy to heterogeneous nodules. In addition, these methods also need to introduce user interaction, prior information, post-processing, or other operations. The limitations of traditional lung nodule segmentation methods indicate that a novel method is needed for accurate and robust lung nodule segmentation.

With the development of convolutional neural networks (CNNs) in computer vision [6,7,8], its application in medical image segmentation has become a research hotspot [9,10,11,12]. Although CNN-based methods have achieved great improvements in segmenting lung tumors compared with traditional approaches [13,14,15], the segmentation of heterogeneous nodules still requires further attention for the following reasons: (1) Large variation between nodules and small inter-class variances between nodules and their surrounding tissues. Nodules come in different types, sizes, locations, etc. (Figure 1a). The intensity may be heterogeneous even within the same nodule (calcific and non-calcific tissues in partially calcified nodules). The intensities of the juxta-pleural, juxta-vascular, and ground-glass opacity (GGO) nodules are indistinguishable from their surrounding lung wall, blood vessels, and lung parenchyma, respectively (Figure 1b). These characteristics hinder their accurate identification. (2) No comprehensive analysis of factors in clinical CT images. Although the location, region, and edge of nodules are three key factors for the diagnosis of lung nodules in practice, researchers usually neglected to consider all of these elements together in segmentation.

As mentioned above, how to improve CNNs approaches to accurately and robustly segment heterogeneous nodules, especially hard-to-segment nodules, is our motivation. Inspired by the diagnosis process of a pulmonary nodule: clinicians first roughly locate a suspicious nodule, then extract the coarse nodule area and accurate nodule edge information according to the local manifestations of the nodule, and finally identify nodules by combining these three factors for further diagnosis and treatment plan. Therefore, we plan to consider building a general segmentation network that combines lung nodule location, coarse region, and edge information. Fortunately, Fan et al. [16] proposed an Inf-net that can segment the COVID-19 infection regions. Although the performance of its network is perfect, it has not been applied to the nodule segmentation task, and it also has the disadvantage of not comprehensively considering the factors in clinical CT images.

To this end, we aim at constructing a general lung nodule segmentation network, namely a features complementary network, which combines the location, coarse area, and edge of lung nodules to achieve coarse-to-fine segmentation. The main contributions of this study are summarized as follows:(1)We propose a novel end-to-end lung nodule segmentation guidance network by fully integrating the global context and spatial information from different scale features, which leverages the complementary information extracted at both small- and large-scales.(2)Under the guidance of assigning more weight to the pixels located at edges and explicitly modeling edges in depth supervision, the location and coarse area are complemented with edge information, which effectively boosts the accuracy and robustness of the lung nodule segmentation model.(3)Experimental results illustrate that the proposed model outperforms other CNNs methods with high accuracy and robustness in lung nodule segmentation performance.

## 2. Related Work

The lung nodule segmentation techniques include traditional image processing-based methods and machine learning-based methods. Traditional techniques include morphology, region growing, level set, and graph cut methods. Machine learning methods can be divided into traditional machine learning methods and deep learning methods, both of which convert segmentation into pixel-classification tasks.

In the morphology method [17], the attached vascular components were first separated by an opening operation, followed by a connected component analysis to retain the nodule volume. The region growing method [18] performed an adaptive sphericity-oriented contrast region growing to distinguish nodules from the lung wall. In the active contour model method, the images were represented as level-set functions [19]. A segmentation of pulmonary nodules study [20] adopted a graph-cut method based on graph theory. However, a common shortcoming of these methods is that one method performs well regarding a certain type of nodule and often performs poorly regarding another. In addition to weak generalization, they often need to add user interaction, prior information, and so on, which is dependent on the human experience.

Recently, the application of machine learning in the segmentation of lung nodules has been ubiquitous. Gonçalves et al. [21] proposed a multi-scale segmentation process for lung nodules based on the Hessian strategy. Mukhopadhyay et al. [22] constructed a two-stage segmentation framework for pulmonary nodules based on internal texture and external attachment features. In addition, a segmentation method that can extract solid and non-solid components in GGO nodules was proposed [23]. Although these traditional machine learning algorithms have achieved excellent accuracy in nodule segmentation, they have encountered some drawbacks, including but not limited to relying highly on manually defined features, being time-consuming, and having weak generalization, which hinder the further development of lung nodule segmentation schemes.

In recent years, deep learning technology has developed rapidly, and CNNs have been widely used for lung nodule segmentation with promising results. Jiang et al. [24] improved the full-resolution residual neural network (FRRN) and designed two lung nodule segmentation networks that combined features of all levels through residual flows. Wang et al. [25] proposed a two-branch structure and used multi-scale for lung nodule segmentation, which is a centrally focused convolutional neural network that combines 3D and multi-scale 2D features. A parallel structure was applied by Cao et al. [26], who also devised a weighted sampling strategy based on nodule boundaries. Similarly, a multi-view CNN with a three-branch structure was adopted by Wang et al. [27] that was fed a set of multi-scale 2D patches from three orthogonal directions: axial, coronal, and sagittal views. Although the parallel structure could effectively integrate multiple features, the model complexity is relatively high, which requires more run-time to reach convergence and increases the risk of overfitting. Especially, Hu et al. [28] paralleled a hybrid attention mechanism with a densely connected convolutional network to segment glioblastoma tumors in an entire lung CT. Their approach is more suitable for larger nodules (glioblastoma diameter range 40–90 mm). In addition to the aforementioned multi-branch parallel architecture, the researchers also designed some other structures. For example, [29,30] are based on CNN and combined with other methods in the pre-processing stage. A hybrid deep learning model [29] applied the adaptive median filter in the pre-processing stage and then used the U-Net-based architecture to segment the lung tumor. In the 2D-3D cascaded CNN framework [30], the CT scan volume was also pre-processed by the maximum intensity projection technique, and then the pulmonary nodules were segmented by the U-Net network which integrated the residual blocks, the squeeze, and excitation blocks. In particular, Song et al. [31] introduced a Faster-CNN model into a generative adversarial network to automatically segment various types of pulmonary nodules. Ni et al. [13] designed a two-stage segmentation algorithm for pulmonary nodules from coarse to fine, which included two multi-scale U-Nets, one of which was used for localization and the other for refinement segmentation. Zhao et al. [14] proposed an improved pyramid deconvolution neural network that fused low-level fine-grained features with finely segmented lung nodules in CT slices. Huang et al. [32] proposed a system for the fully automatic segmentation of lung nodules directly from raw thoracic CT scans. Although [14,32] improved the segmentation accuracy by fusing all low-level features, the computational burden was increased because of integrating low-level features equally, while not fully reusing high-level features.

Furthermore, deep learning methods focusing on multi-scale have also been applied to lung nodule segmentation. For example, Maqsood et al. [33] proposed a U-Net-based segmentation framework that integrates dense deep blocks and dense Atrous blocks. Shi et al. [34] presented a lung nodule segmentation model multi-scale residual U-Net (MCA-ResUNet), which applies Atrous Spatial Pyramid Pooling (ASPP) as a bridging module and adds three adjacent smaller-scale guided Layer-crossed Context Attention (LCA) mechanisms. A semi-supervised three-view segmentation network with detection branches was proposed by Sun et al. [35], but three parallel dilated convolutions for multi-scale feature extraction were performed in the detection and classification modules. Based on the encoder-decoder model, Wang et al. [36] changed skip connections to multiple long and short skip connections. In addition, a global attention unit and a boundary loss were added to segment difficult-to-segment (DTS) nodules. Through skip connections, each convolutional block of the decoder can access its feature maps of each previous layer at the same level to aggregate multi-scale semantic information. Yang et al. [37] used a ResNet structure to improve 3D U-Net, which focuses on adopting deep supervision to guide the network to extract multi-scale features, rather than fusing features at different scales. Specifically, it adds side-depth supervision to each layer in the decoder. Considering the complementarity between the nodule patch and the global CT, Wang et al. [38] proposed a dual-branch multigranularity scale-aware network (MGSA-Net), which unifies the representation of global- and patch-level image features in one framework. The deep scale-aware module (DSAM) in the global branch extracts the concealed multi-scale contextual information at different stages through three parallel branches. Ni et al. [13] constructed a two-stage network for lung nodule segmentation and classification. A 3D multi-scale U-Net (MU-Net) was employed in the first stage to locate nodules. In the second stage, a 2.5D multiscale separable U-Net (MSU-Net) adopts a multi-branch separable convolutional input layer to extract features of different scales from any input image scale to refine the output of MU-Net. Similarly, the models of Wang et al. [38] and Ni et al. [13] are all improved on the basis of U-Net. Studies by Yang et al. [37] and Ni et al. [13] are 3D networks that exploit the continuity of information between CT slices. In general, 3D networks have more parameters than 2D networks, which may easily lead to overfitting and slow convergence speed. Especially if there is a lack of enough labeled samples during training, the performance of the 3D network is worse than that of the 2D network. Zhu et al. [39] added a High-Resolution network with Multi-scale Progressive Fusion (HR-MPF) in the encoder part of the High-Resolution Network (HRNet) and proposed a Progressive Decoding Module (PDM) in the decoder part. In addition, a loss function with edge consistency constraint is designed in the segmentation loss. 

Note that there are two common shortcomings in all the above-mentioned studies that use CT patches for lung nodule segmentation. First, the tumor-centered CT patches were used uniformly, which meant that the location of the tumor in the CT patch was fixed. Therefore, the segmentation performance was likely to be biased as it did not depend on the position of the tumor in the raw CT slice. The other issue was the use of a fixed-size square CT patch for feature extraction, which may contradict the large changes in the sizes and shapes of lung nodules.

Compared to the previously developed CNNs, our model differs in the following: (1) Our model multi-scales the input 2D CT patches and further cross-scale weighted aggregates high-level multi-scale features to extract global features containing rich location and semantic information when only one network is involved. (2) Considering that low-level and high-level features are different, the model does not integrate them equally; instead extracts them separately in a manner that ensures they complement each other, which reduces the calculation complexity. (3) The edge information is explicitly modeled to preserve the nodule boundaries. In particular, the nodule location information is introduced into the edge information to strengthen the edge features. (4) The tumor location and size of the CT patches in the dataset are not fixed, which improves the robustness of the model.

The remainder of this paper proceeds as follows. Section 3 describes the proposed method in detail. The datasets and experimental details are presented in Section 4. Section 5 presents a comparison between the qualitative and quantitative experimental results. Finally, we discuss some potential improvements and draw conclusions in Section 6.

## 3. Materials and Methods

Figure 2 illustrates the architecture of the proposed network, including four major parts: the backbone, HDM, LDM, and CM. Our proposed model uses pre-trained Res2Net50 as the backbone and takes CT patches of three scales to capture coarse multi-scale features. To enhance the representation ability of the model and adapt to the segmentation task setting, we replace a 7 × 7 convolution in layer 1 with three consecutive 3 × 3 convolution and ReLU layers and remove the last pooling layer and fully connected layer. As such, layer 1 and layer 2 with low-level features contain rich edge information, while layer 3, layer 4, and layer 5 with high-level features embrace strong semantic information. Next, HDM takes high-level rough multi-scale features as its input could acquire refined multi-scale features with rich spatial information whilst suppressing irrelevant background noise. HDM extracts the nodule location and the coarse area of different size nodules. Meanwhile, the high-resolution low-level edge information is fed into the LDM to obtain initial edge information on the basis of reducing computer memory. Finally, CM is used to perform complementation of the location, coarse area, and edge of a lung nodule. Concretely, CM makes location information supplement on initial edge information through location fusion (LF), and refined edge information supplement on coarse nodule area via edge fusion (EF). 

### 3.1. High-Level Feature Decoder Module (HDM)

Researchers apply different sizes of convolution kernels to obtain multi-size receptive fields, which are designed to be superior to those that share a fixed size. Here, we design an MF block to capture more spatial information on nodules of different sizes by four cascade branches {b_m_, m = 0,…,3}, which is inspired by the receptive field block (RFB) [40]. As shown in Figure 3, each branch consists of a standard and dilated convolutional layer. As the convolution kernel size and the atrous convolution dilation rate of the four branch increases from 1 to 3, 5, and 7, then the receptive fields will be 1, 9, 15, and 21, respectively. To be specific, every branch first applies a 1 × 1 convolutional layer to reduce the number of channels to 32. To further reduce the number of parameters and deeper non-linear layers, for {bm, m ≥ 1}, we replace (2 m + 1) × (2 m + 1) convolutional layers with a 1 × (2 m + 1) and a (2 m + 1) × 1 convolutional layers, followed by a 3 × 3 convolutional layer with a (2 m + 1) dilation rate, which is widely used in Deeplab [41]. We then concatenate the output of the abovementioned four branches and send it to a 3 × 3 convolutional layer to reduce the number of channels from 4 × 32 to 32. Finally, a shortcut is added elementwise to the original MF block. To extract richer high-level semantic features and reserve more spatial information, we added three MF blocks to the HDM. In particular, the MF block we added in layer 4 and layer 5 can further compensate for the loss of spatial information and capture a more accurate position of a nodule, which can be used as a feature supplement for the LE block.

Many segmentation tasks consider all high- and low-level features of the backbone equally and aggregate them uniformly [42]. However, compared to high-level features, low-level features have higher resolution and contain weaker semantic information, which require more computation cost and contribute less to the segmentation results [43]. Based on the aforementioned reasons, we designed an MD block, as shown in Figure 4, which only progressively integrates three high-level features. Specifically, for a 2D CT image, we first extract two low-level features $\left\{f_{i},i=1,2\right\}$ and three high-level features $\left\{f_{i},i=3,4,5\right\}$ through five convolution layers of Res2Net. We then apply the MD block to gradually integrate high-level features and generate the coarse area of pulmonary nodules $f_{g}$, which will continue to be supplemented by location and edge information in subsequent CM. We set l=3 and L=3, the MD block operation is defined as follows:(1)$$f_{g}=f_{3}^{\prime \prime }©f_{4}^{\prime \prime }©f_{5}^{\prime }$$
where $f_{g}$ is the initial aggregated global coarse nodule area, © denotes the concatenating operation, and $f_{i}^{\prime }$ and $f_{i}^{\prime \prime }$ are the multi-scale context feature output by the MF block and its corresponding updated feature, respectively. For the deepest feature i=L, we set $f_{i}^{\prime \prime }=f_{i}^{\prime }$.

For the updated feature map $\left\{f_{i}^{\prime \prime },i\in \left[l,\ldots ,L-1\right]\right\}$, which is obtained by multiplying its original feature with the remaining deeper feature maps, $f_{i}^{\prime \prime }$ is defined as follows:(2)$$f_{i}^{\prime \prime }=f_{i}^{\prime }⊗\prod_{k=i+1}^{L}Conv\left(Up\left(f_{k}^{\prime };f_{i}^{\prime }\right)\right),\;i\in \left[l,\ldots ,L-1\right]$$
where $Up\left(f_{k}^{\prime };f_{i}^{\prime }\right)$ is the up-sampling operation that aims to resize $f_{k}^{\prime }$ to the same size as $f_{i}^{\prime }$ by bilinear interpolation, Conv is a 3 × 3 convolutional layer with BN, and ⊗ means multiplication in an element-wise manner.

Finally, we obtain a progressive aggregated feature map with two 3 × 3 and one 1 × 1 convolutional layer, which is a coarse nodule area. The aggregation method of the MD block fully reuses the high-level global features through weighted cross-scale integration.

### 3.2. Low-Level Feature Decoder Module (LDM)

It is known that in the stage of down-sampling feature extraction (Res2Net50 in this paper), low-level feature maps attain significant high-resolution edge information [43]. Meanwhile, many researchers have pointed out that edge information can be used as an a priori constraint, which can effectively improve segmentation performance [10,16,44]. The edge of pulmonary nodules is also one of the key pieces of information that clinicians pay attention to in clinical diagnosis. Therefore, we must consider that the lower-level features ($f_{1}$ and $f_{2}$ in our model) retain enough edge information. We input these edge features into the proposed LE block (Figure 5) to yield a complete edge feature map $f_{e}$ with moderate resolution. Specifically, layer 1 extracts local edge features $f_{1}$, while layer 2 captures more abstract global edge features $f_{2}$; the two low-level edge features can complement and enhance each other in a positive manner. The workflow is as follows: two shallow features {$f_{1},f_{2}\}$ are first sent to a set of filters that can capture a robust edge feature map. Then, they are fused to produce an original edge feature map $f_{E}$. The LE block function could be represented by the following:(3)$$f_{E}=f_{2}⊕\left(f_{1}⊗f_{2}\right)$$
where ⊕ and ⊗ are the addition and multiplication operations in an element-wise manner, respectively. Unlike ⊕, which emphasizes complementary features, ⊗ emphasizes the enhancement of common features.

### 3.3. Complementary Module (CM)

This module includes location fusion (LF) and edge fusion (EF) (see the upper right of Figure 2), which aim to complement and enhance the location and edge information while extracting enhanced edge features and explicitly modeling the enhanced edge information. More prominent edge features can be obtained by introducing high-level semantic information or location information into the local edge information [45]. Inspired by [46,47], we take $f_{4}^{\prime }$ and $f_{5}^{\prime }$ obtained from the MF block as more accurate location information, and then combine them with the original edge information $f_{E}$ to obtain the final edge guidance information $\hat{f_{E}}$, which can effectively constrain the edge in segmentation. After obtaining the final edge guidance information and coarse nodule area, we utilize the final edge guidance information to further refine the coarse nodule area to achieve fine segmentation. Specifically, the LF block first adjusts the size of $f_{5}^{\prime }$ to be consistent with $f_{4}^{\prime }$ through a set of convolutions and up-sampling, and then it is dotted with $f_{4}^{\prime }$ to obtain an accurate location feature map. The final edge feature map $\hat{f_{E}}$ is obtained by adding the original boundary information $f_{E}$ point-by-point to explicitly learn the edge representation of the lung nodule. Finally, we use EF block to combine the final nodule edge $\hat{f_{E}}$ with the coarse area of pulmonary nodule $f_{g}$ through convolution, up-sampling, and addition operation to obtain the final nodule segmentation prediction map $P_{s}$. The location fusion (LF) and edge fusion (EF) operations are expressed as follows:(4)$$\hat{f_{E}}=Conv(f_{E}⊕Up(f_{4}^{\prime }⊗⸹\left(Up\left(Conv\left(Convs\left(f_{5}^{\prime }\right)\right)\right);f_{4}^{\prime }\right);f_{E}))$$
(5)$$P_{s}=\hat{f_{E}}⊕Up\left(\left(Conv\left(Convs\left(f_{g}\right)\right)\right);\hat{f_{E}}\right)$$
where *Convs* is a set of convolutional operations that aims to capture features with rich detailed information. Furthermore, *Conv* is a convolutional operation with a ReLU activation function that can change the number of channels. $Up\left(*;f\right)$ is the up-sampling operation, which is used to resize ∗ to the same size of f by bilinear interpolation, ⸹ represents a sigmoid function, ⊗ and ⊕ denote element-wise multiplication and summation.

Here, to explicitly model the enhanced edge features, we add an extra edge supervision that measures the difference between the final predicted edge map $\hat{f_{E}}$ and edge map $G_{e}$ generated by the groundtruth (GT). We use the standard binary cross entropy (BCE) loss $L_{edge}$ as an edge constraint:(6)$$L_{edge}=-\overset{M}{\underset{i}{\sum }}\left[G_{e}log\left(\hat{f_{E}}\right)+\left(1-G_{e}\right)\log\left(1-\hat{f_{E}}\right)\right]$$
where M denotes the total number of pixels and $G_{e}$ is the edge groundtruth map, which is obtained by calculating the gradient of the groundtruth map $G_{s}$. Equation (6) is the edge supervision that we added to supervise the edge feature map.

### 3.4. Loss Function

Inspired by reference [48], we consider that pixels located at edges contain more texture information than other pixels, so we pay more attention to edges during segmentation. In this paper, each pixel is assigned a weight W. The edge pixel corresponds to a larger W, while the non-edge pixel corresponds to a smaller one, and W can be used as an indicator of pixel importance. $W_{i}$ denotes the weight map, which is derived from the ground truth map $G_{s}$, and is calculated as follows:(7)$$W_{i}=\alpha +\beta \left|G_{S}-\frac{1}{n}\underset{i\in A_{n}}{\sum }G_{S}\right|$$
where $A_{n}$ denotes the area that surrounds the pixel i. α and β are the threshold and intensity parameters, respectively, which are hyperparameters. Here, we empirically set α=1 and β=5. In summary, we extracted pixels located at edges through average pooling and subtraction operations. $W_{i}$ has the same size as the groundtruth map $G_{s}$. When the pixel in question is located at the edges, it will be assigned a large weight, and vice versa. Figure 6 visualizes the weight distribution of nodules using this weighting strategy, which explicitly weighs more edges into the segmentation loss.

Based on the above analysis, to obtain better segmentation performance, in addition to the edge loss $L_{edge}$ for edge supervision proposed in Equation (6), we propose the joint segmentation loss $L_{seg}^{\omega }$, which is used for deep segmentation supervision.
(8)$$L_{seg}^{\omega }=\frac{1}{2}\left(L_{BCE}^{\omega }+L_{IOU}^{\omega }\right)$$
where $L_{BCE}^{\omega }$ and $L_{IOU}^{\omega }$ are the weighted BCE and weighted IOU loss, respectively. $L_{BCE}^{\omega }$ is a pixel-level loss defined as follows:(9)$$L_{BCE}^{\omega }=\frac{-\sum_{i}^{M}W_{i}*\left[G_{S}log\left(P_{s}\right)+\left(1-G_{S}\right)\log\left(1-P_{s}\right)\right]}{\sum_{i}^{M}W_{i}}$$
where $P_{s}$ is the predicted global map. In contrast to the standard BCE loss, the weighted BCE loss $L_{BCE}^{\omega }$ assigns a larger effect coefficient to the pixels located at edges to increase their loss contribution to the loss. Considering that $L_{BCE}^{\omega }$ ignores the global structure of the image when calculating the loss of each pixel independently, we introduce weighted IOU loss $L_{IOU}^{\omega }$ to make the network pay more attention to the global structure. $L_{IOU}^{\omega }$ is image-level loss, which is widely used in segmentation and object detection. It is designed to optimize the global structure rather than focusing on a single pixel, which can be used as a complement to $L_{BCE}^{\omega }$. Similar to $L_{BCE}^{\omega }$, the weighted IOU loss $L_{IOU}^{\omega }$ focus on pixels located at edges through the following weighting method:(10)$$L_{IOU}^{\omega }=1-\frac{\sum_{i}^{M}W_{i}*\left(P_{s}*G_{S}\right)}{\sum_{i}^{M}W_{i}*\left[P_{s}+G_{S}-P_{s}*G_{S}\right]}$$

As shown in Equation (8), the segmentation loss function includes both local and global losses, which complement each other and provide effective supervision for accurate lung nodule segmentation.

The total loss function of the proposed network consists of two parts: one part tackles the most common segmentation supervision presented in Equation (8), and the other focuses on the edge supervision described in Equation (6), which plays a crucial role in medical segmentation. Therefore, the total hybrid loss is defined as:(11)$$L_{total}=L_{edge}+L_{seg}^{\omega }$$

## 4. Experiments

### 4.1. Datasets

To evaluate the performance of the proposed network, we conducted experiments on two datasets. One is a public benchmark dataset: the LUng Nodule Analysis 2016 (LUNA16) dataset [49], and the other is an independently collected dataset from the Fudan University Shanghai Cancer Center (FUSCC). 

**LUNA dataset:** There are 888 CT scans and 1186 GT nodules in LUNA16, which exclude slices thicker than 2.5 mm obtained from the Lung Image Database Consortium and Image Database Resource Initiative (LIDC-IDRI) [50]. The LIDC-IDRI database contains annotations collected through a two-stage phase annotation process by four experienced radiologists. Among all the marked lesions, only nodules ≥ 3 mm and accepted by at least three out of four radiologists constitute the LUNA16 dataset. In other words, annotations that do not conform to the reference standard (non-nodules, nodules < 3 mm, and nodules annotated by only one or two radiologists) are referred to as irrelevant findings.

**FUSCC dataset:** The second dataset contains 1134 CT slices of nodules from 89 subjects with single nodules admitted at the Fudan University Shanghai Cancer Center. All nodules are randomly assigned to four board-certified radiologists for labeling, then, verified and corrected by an experienced radiologist (10+ years of experience). Generally, each nodule has several to dozens of slices in raw CT volume, and we take each slice as a sample.

### 4.2. Evaluation Metrics

To qualitatively evaluate the segmentation performance, we employ five general evaluation metrics, namely, Dice similarity coefficient (DSC), Jaccard Index (JA), Hausdorff distance (95%) (HD95), specificity (SP), sensitivity (SE). In addition, three gold standard evaluation metrics in the salient object detection field have been introduced, i.e., S-measure (Sm), E-measure (Em), and mean absolute error (MAE). 

Dice similarity coefficient (DSC) and Jaccard Index (JA) are common evaluation criteria that are used to calculate the overlap ratio between the segmentation result (S) and ground-truth (GT). They both range from 0 to 1, where 1 means perfect overlap [51]. It is calculated as below:(12)$$DSC=\frac{2TP}{2TP+FP+FN}=\frac{2\left|S\cap GT\right|}{\left|S\right|+\left|GT\right|}$$
(13)$$JA=\frac{TP}{TP+FN+FP}=\frac{\left|S\cap GT\right|}{\left|S\cup GT\right|}=\frac{\left|S\cap GT\right|}{\left|S\right|+\left|G\right|-\left|S\cap GT\right|}$$
where *TP*, *FP*, and *FN* represent the number of true positives, false positives, and false negatives, respectively. | · | refers to calculating the number of pixels in a given region. ∩ and ∪ denote taking the intersection and union, respectively.

Hausdorff distance (HD) is formulated as:(14)$$HD\;\left(S,GT\right)=max\left\{max_{p\epsilon S}\left\{min_{t\epsilon GT}\left|\left|p-t\right|\right|\right\},max_{t\epsilon GT}\left\{min_{p\epsilon S}\left|\left|t-p\right|\right|\right\}\right\}$$
where p and t are pixels on S and GT, respectively. As suggested in [52], we use the Hausdorff distance (95%) (HD95) to eliminate the adverse effects of outliers.

S-measure (*Sm*) computes the structural similarity between the segmentation result and ground-truth, which is defined as follows:(15)$$S_{m}=\left(1-\alpha \right)\cdot S_{o}\left(S,GT\right)+\alpha \cdot S_{r}\left(S,GT\right)$$
where α is the balance coefficient between object-aware similarity $S_{o}$ and region-aware similarity $S_{r}$, which is set to 0.5.

E-measure (Em) jointly evaluates the local and global similarities between the binarized prediction and ground truth, which is defined as:(16)$$E_{m}=\frac{1}{M}\overset{M}{\underset{i}{\sum }}\emptyset \left(S_{i},GT_{i}\right)$$
where i and M mean each pixel and the total number of pixels in the GT, respectively. The symbol ∅ indicates the enhanced alignment matrix.

MAE reflects the pixel-wise error between S and GT, which is denoted as follows [53]:(17)$$MAE=\frac{1}{M}\overset{M}{\underset{i}{\sum }}\left|S_{i}-GT_{i}\right|$$

### 4.3. Implementation Details

We divide the two datasets into training and testing sets at a patient level with the same ratio of 9:1 before the experiments. At the beginning of training, we first resize all patches or regions of interest (ROIs) to 96 × 96 and then multi-scale the input patches. Specifically, the network applies bilinear interpolation to resample the input single patch with three ratios of 0.75, 1, 1.25 to obtain three scale images. In other words, our model is trained with a multi-scale strategy. Note that in order to obtain the different locations of the same nodule in patches, we perform five cropping operations around the same nodule on the same CT slice and the size of the patch is not a fixed square.

The entire framework is implemented with Python3.6 based on PyTorch-GPU 1.4.0 on an experimental platform consisting of an Ubuntu 18.04 operating system with an NVIDIA GeForce GTX 1080 Ti graphics cards and 32 Gigabytes of memory. We adopt the Adam optimizer for training with an initial learning rate of 1e-4 that dropped 10% every 30 epochs. To avoid overtraining, if the performance stabilizes, training is stopped after ten extra epochs. We found that our model converges after approximately 50 epochs with a batch size of four. Consequently, we set the upper limit of the training period to 60 epochs. The performance of our approach was evaluated using MATLAB 2018a.

## 5. Results

### 5.1. Quantitative Analysis

To verify the efficiency of the proposed model, we performed a quantitative comparison with the SOTA methods on the LUNA16 and FUSCC test datasets, namely residual U-Net (RUN) [54], Huang et al. [32], U-Net++ [55], U-Net [42], CE-Net [56] and Attention U-Net [57]. Table 1 reports the overall segmentation performance of all methods and datasets based on multiple indicators. We observe that our proposed model outperforms almost all methods on all evaluation metrics in the LUNA16 dataset. In addition, when tested on an independent FUSCC dataset, the similar good experimental results of our proposed model reaffirm the competitiveness for the segmentation of different types of pulmonary nodules. Our-Net obtains significantly better DSC, JA, and MAE than U-Net and U-Net++ for all data sets. Specifically, our network improves from 0.676, 0.511, 0.121 for U-Net and 0.747, 0.596, 0.087 for U-Net++ to 0.835, 0.717, 0.015 for DSC, JA, and MAE indexes on LUNA16, while it optimizes from 0.686, 0.522, 0.137 for U-Net and 0.767, 0.621,0.091 for U-Net++ to 0.868, 0.767,0.026 for DSC, JA, and MAE indicators on FUSCC dataset. Furthermore, our model yields lower Hausdorff distances (95%) (HD95) than all comparison methods in all databases. The SP, Sm, and Em indicators of the proposed network are all slightly improved compared with the comparison methods. In addition, our network achieves 0.835, 0.717, and 0.015 in DSC, JA, and MAE measures with limited improvement, while the second-best values of 0.825, 0.702, and 0.019, respectively. This shows that the classical CE-Net proposed to capture more high-level information and spatial information is very effective for segmentation. Particularly, although Attention U-Net obtains the highest SE value (0.907), its SP and DSC values are relatively lower (0.982 and 0.772, respectively). The combination of the three metrics illustrates that Attention U-Net is more prone to mis-segmentation compared to our method, which is not conducive to clinical diagnosis. Our approach is statistical significance compared to U-Net, U-Net++, and Attention U-Net (*p*-value < 0.05). The superiority of our network in lung nodule segmentation may be owed to the complementarity of the sufficient refinement feature and weighted cross-scale fusion feature.

### 5.2. Ablation Studies

To verify the effectiveness of each component in our model, we conducted a series of ablation experiments on the FUSCC test dataset, as shown in Table 2. We observed that the results of almost all evaluation metrics increased as the components were added sequentially. In particular, we demonstrate the huge advantage of the proposed MF block by comparing row (a) with row (b). Its application boosts the DSC value by a substantial 44% (from 0.459 to 0.659) and the HD95 value reduces from 34.703 to 26.286. This indicates that MF blocks can extract features of different size nodules by combining atrous convolution with different atrous rates. We also compare row (b) with row (c), for example, JA increases from 0.491 to 0.654 by 33%, HD95 decreases from 26.286 to 8.389 while MAE decreases from 0.101 to 0.076, which further supports that our proposed MD block is beneficial for pulmonary nodule segmentation. Finally, we demonstrate the effectiveness of the proposed weighted strategy for pixels located at edges by comparing the fusion result with and without the edge weighting (row (d) and row (e)) in CM, where (w/o) represents our model without the edge weighting strategy. Table 2 manifests that the edge pixel weighting method boosts DSC by about 8% on FUSCC. Specifically, our proposed network achieves good performance on the FUSCC (DSC: 0.868, JA: 0.767, HD95: 5.354, SE: 0.884, SP: 0.987, Sm: 0.919, Em: 0.962, and MAE: 0.026). Overall, the evaluation metrics in Table 2 are all further improved, which confirms that the proposed components are effective in learning pulmonary nodule features. Moreover, we also make a visual analysis of the feature learning ability of each component in our model as shown in Figure 7. As can be observed, the segmentation edges gradually approach the ground-truths, and the final segmentation edges are closest to the ground-truths, which indicates that OUR-Net is effective in locating nodule edges.

In addition, in order to explore the complementary performance of location information in our network, we sequentially added the location information $f_{3}^{\prime }$, $f_{4}^{\prime }$, and $f_{5}^{\prime }$ obtained from layer 3, layer 4, and layer 5 to CM on the FUSCC test set, as shown in Table 3. It can be seen that the addition of pulmonary nodule location information significantly boosts the segmentation performance, where the combination of $f_{4}^{\prime }$ and $f_{5}^{\prime }$ obtains the optimal values of 0.868, 0.767, and 5.354 on DSC, JA, and HD metrics, respectively. Its performance is better than the single $f_{5}^{\prime }$ or the combination of $f_{3}^{\prime }$, $f_{4}^{\prime }$ and $f_{5}^{\prime }$, which demonstrates $f_{4}^{\prime }+f_{5}^{\prime }$ jointly provides enough and the most balanced location information.

### 5.3. Qualitative Analysis

Figure 8 displays representative nodule segmentation edges from the FUSCC and LUNA testing sets (F1–F5 and L1–L5, respectively) to visually compare our approach to other approaches; including U-Net, U-Net++, and Attention U-Net. Specifically, for spiculate nodules (F1) found on the FUSCC, U-Net excessively segments the surrounding tissues. When segmenting the juxta-pleural nodules (F2), it is arduous to distinguish nodules and surrounding tissues with the same strength using U-Net and Attention U-Net, and U-Net++ segments the nearby pleura of similar intensity excessively. Regarding GGO nodules (F3), owing to the low contrast, Attention U-Net excessively segments the lung parenchyma. When segmenting isolated nodules (F4), U-Net falsely segments nearby tissues. In cavitary nodules (F5) segmentation, U-Net++ segments nodules only partially owing to low contrast. U-Net and Attention U-Net struggle to distinguish nodules from complex surroundings. In contrast, our network maintains strong robustness when segmenting these types of nodules. For simplicity, we analyzed typical nodule segmentation results on the LUNA dataset. When U-Net and Attention U-Net attempt to segment calcified nodules (L2), they cannot distinguish the background. When segmenting the GGO nodules with cavity structures (L4), U-Net and U-Net++ only partially recognize the nodules owing to the low-intensity contrast. Attention U-Net is slightly aggressive in segmentation, while U-Net++ is slightly conservative. From the above-mentioned qualitative comparison, it could be observed that the challenging nodules are mostly juxta-pleural nodules, juxta-vascular nodules, and nodules with heterogeneous intensities. Figure 9 further visualizes the segmentation results of these types of nodules from the LUNA and FUSCC datasets using our method. Notice that the segmentation results of our model have a large overlap with the ground-truth, which shows that it can always obtain the most accurate segmentation edge in the trade-off. The strong robustness of our method may benefit from the combination of the proposed components that are surely helpful for the refinement and fusion of complementary information between cross-scale features. Visual qualitative experiments show that our approach is effective in the segmentation of various types of lung nodules.

Unexpectedly, we found a wrong GT label for a juxta-vascular nodule in the comparative examination of test results (as shown in Figure 9, L13). Fortunately, our proposed network still accurately segmented it without being affected by the erroneous GT label, which proved the robustness and consistency of our model in pulmonary nodule segmentation.

### 5.4. Discussion

With the development of computer hardware technology and deep learning algorithms, more and more convolution neural networks are designed for the automated analysis of medical images. Although deep learning models have achieved marvelous results in various medical image tasks, precision lung nodule segmentation remains a challenging task owing to the diversity of lung nodules, the blurry edges, and small inter-class variances between nodules and their surrounding tissues. Inspired by the process of clinical diagnosis, we design a model in which the location, region, and edge information complement each other.

The experimental results show that the proposed network can segment pulmonary nodules effectively and robustly. Our MF block focuses on pulmonary nodule areas with rich details and yields locations with rich spatial information. The MD block generates coarse nodule areas by weighted cross-scale feature fusion and suppresses irrelevant information. In addition, our CM refines the nodule edge by complementing each other on location, region, and edge information to achieve the accurate segmentation of pulmonary nodules. These components can support plug-and-play to flexibly and effectively combine with other networks to improve performance. It is worth noting that OUR-Net achieved inspiring performance in pulmonary nodule segmentation in CT patches without any pre- or post-processing tricks.

However, there are still some limitations in our work. Although our approach achieved good results in lung nodule segmentation, it is only for 2D image applications. Additionally, nodule segmentation is only one part of lung cancer analysis and diagnosis. In the future, we will consider introducing the correlated inter-slice information of the 3D volume into the model to obtain better segmentation results. What is more, we will integrate the segmentation and classification of pulmonary nodules into a unified framework, and take advantage of the correlation between tasks to study a detection model in which segmentation and classification promote each other. Therefore, the framework can be better applied to clinical analysis.

## 6. Conclusions

In this study, we propose a novel scheme for lung nodule segmentation, which extracts high- and low-level local features in different ways and complements them with each other. Our proposed network is trained in an end-to-end manner to obtain an effective and robust segmentation performance for different types of pulmonary nodules. Our core idea is to imitate the nodule determination process so that nodule location, coarse nodule area, and nodule edge complement each other in clinical diagnosis, so that the network can learn multi-scale features with high consistency. Compared to several classic lung nodule segmentation methods, our method demonstrates excellent performance (DSC = 0.835 ± 0.002 for LUNA, and DSC = 0.868 ± 0.001 for FUSCC). In particular, our model exhibits great potential in segmenting challenging nodules, such as juxta-pleural nodules, juxta-vascular nodules, and nodules with heterogeneous intensity.

## Figures and Tables

**Figure 1 entropy-24-01755-f001:**
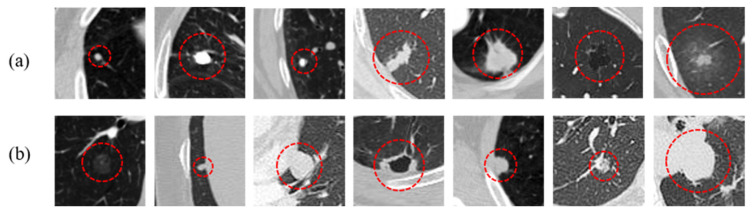
Examples of pulmonary nodules with large variation between nodules and small variation between nodules and their surrounding tissues in CT patches. (**a**) Indicates that nodules are high variability in size, shape, and intensity. (**b**) Indicates that nodules have similar visual characteristics to their surrounding lung parenchyma, lung wall, blood vessels, etc.

**Figure 2 entropy-24-01755-f002:**
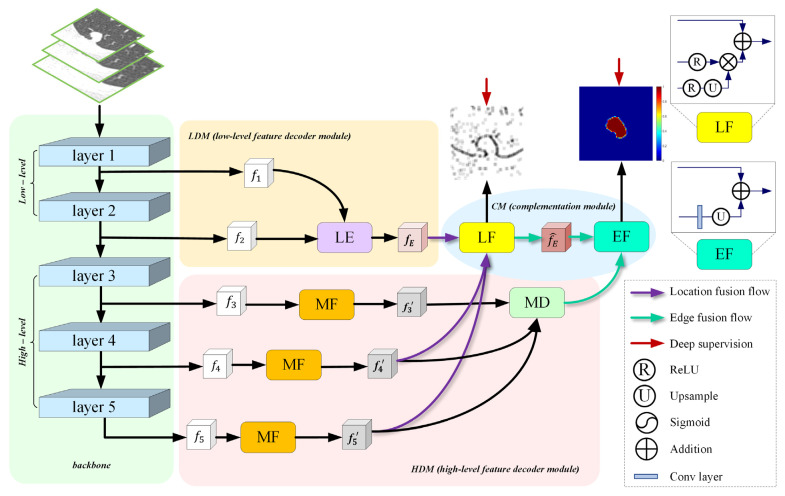
The structure of the proposed network. **backbone**: Res2Net50 that is pre-trained on ImageNet dataset. **HDM**: high-level feature decoder module. It fully aggregates multi-scale high-level features to generate rough global features. **LDM**: low-level feature decoder module. The generated edge features are expected to refine and constrain the coarse global features. **CM**: complementary module. It complements the edge and global features to obtain a segmentation prediction map. **LE**: low-level edge block. **MF**: multi-receptive field block. **MD**: multi-scale decoder block. **LF**: location fusion. **EF**: edge fusion.

**Figure 3 entropy-24-01755-f003:**
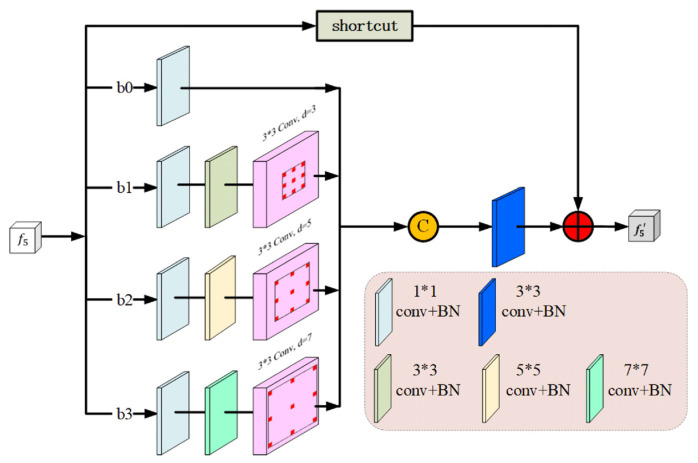
The architecture of the multi-receptive field (MF) block, “d” denotes dilation rate. It contains four branches that can extract features from different scales to retain more accurate spatial location information at high levels.

**Figure 4 entropy-24-01755-f004:**
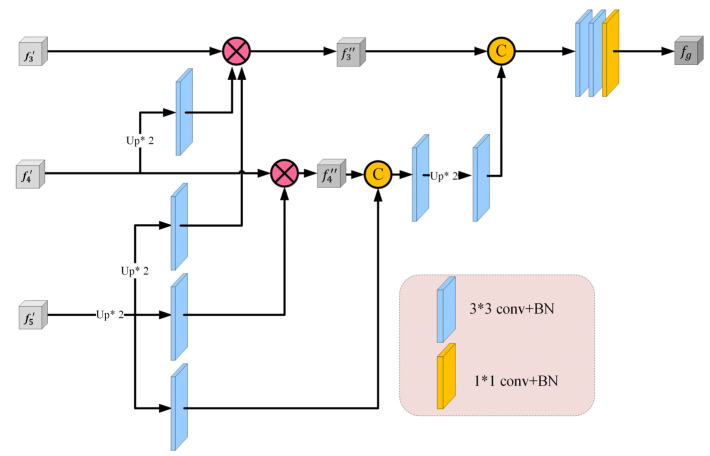
The architecture of the multi-scale decoder (MD) block. It aggregates three high-level features cross-scale to generate a coarse global map of lung nodules, which makes full use of the high-level features.

**Figure 5 entropy-24-01755-f005:**
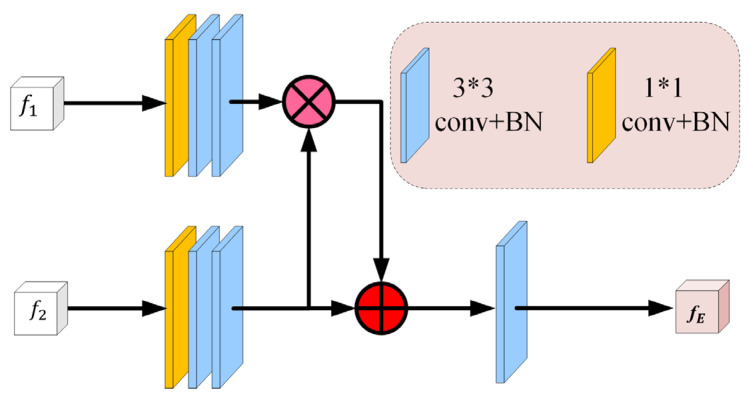
The architecture of the low-level edge (LE) block. It extracts complete edge information, which effectively constrains and guides the lung nodule segmentation, and further refines the coarse lung nodule area.

**Figure 6 entropy-24-01755-f006:**
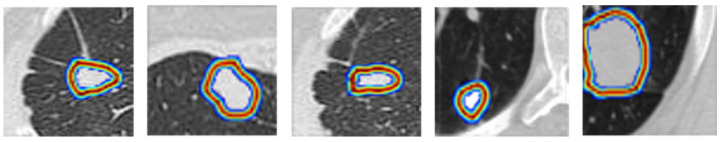
Visualization of weighted pixels located at edges from different nodule samples. The nodule pixel weight distribution under the weighted strategy for pixels located at edges, where red is the high weight and blue is the low weight.

**Figure 7 entropy-24-01755-f007:**
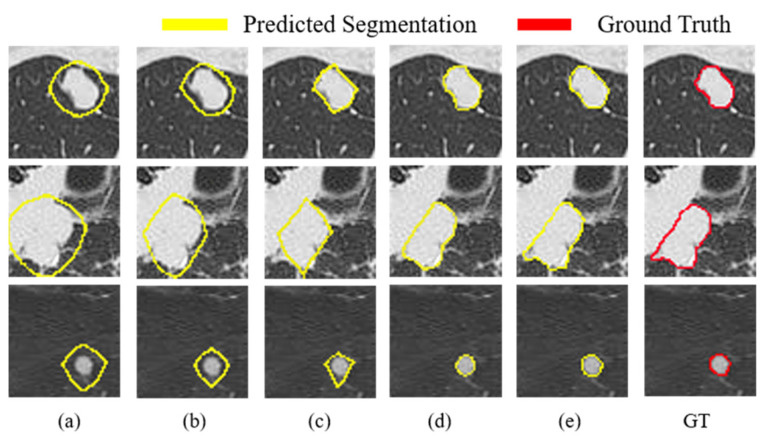
Visual comparisons between the proposed modules on the test set. Each row presents a nodule segmentation case and each column is the segmentation result of adding different components: (**a**) backbone. (**b**) backbone + MF. (**c**) backbone + MF +MD. (**d**) backbone + MF + MD + CM without loss strategy for weighted pixels located at edges. (**e**) backbone + MF + MD + CM with loss strategy for weighted pixels located at edges. GT: Ground-Truth.

**Figure 8 entropy-24-01755-f008:**
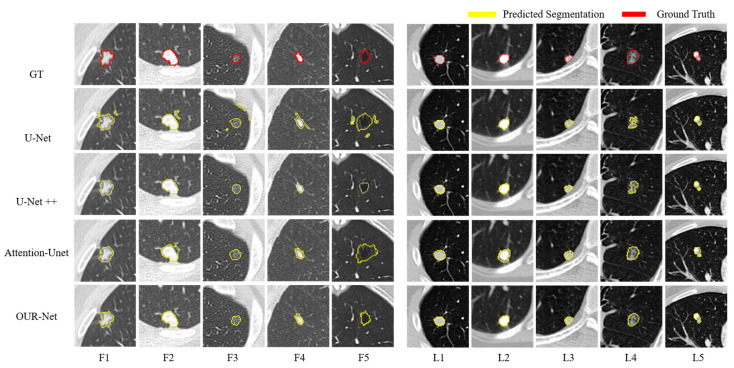
Visualization of lung nodule segmentation results. From top to bottom: the image of GT, segmentation result of U-Net, U-Net++, Attention U-Net, and OUR-Net. F1–F5 and L1–L5 are 5 representative nodules from the FUSCC and LUNA testing sets, respectively.

**Figure 9 entropy-24-01755-f009:**
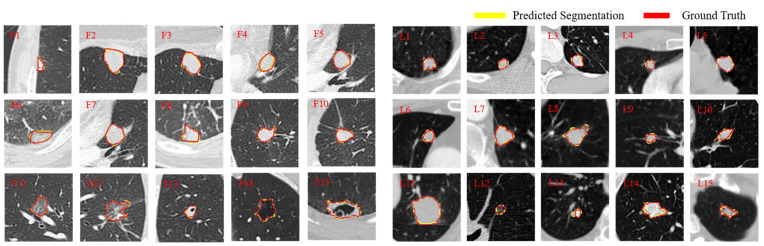
Segmentation results of the challenging nodules by OUR-Net on FUSCC and LUNA testing sets (F1–F15 and L1–L15, respectively). The red and yellow outlines represent the GT and segmentation results of OUR-Net, respectively.

**Table 1 entropy-24-01755-t001:** Performance comparison of lung nodule segmentation on the LUNA16 and FUSCC test datasets. DSC, JA, and HD95 are presented as mean ± standard deviation (SD) (95% confidence interval). The best two results are indicated in **red** and **blue**. **↑:** larger is better, **↓**: smaller is better.

Method		LUNA16						FUSCC				
	U- Net	U-Net++	RUN	Attention U-Net	Huang et al.	CE- Net	OUR-Net	U- Net	U-Net++	Attention U-Net	CE- Net	OUR-Net
**DSC** **↑**	0.676 ± 0.004	0.747 ± 0.003	0.719 N/A	0.772 ± 0.006	0.793 N/A	**0.825** ± 0.002	**0.835** ± 0.002	0.686 ± 0.003	0.767 ± 0.009	0.816 ± 0.006	**0.864 **± 0.003	**0.868** ± 0.001
**JA** **↑**	0.511 ± 0.004	0.596 ± 0.004	0.561 N/A	0.628 ± 0.008	0.657 N/A	**0.702** ± 0.002	**0.717** ± 0.003	0.522 ± 0.003	0.621 ± 0.012	0.675 ± 0.008	**0.761 **± 0.004	**0.767** ± 0.002
**HD95** **↓**	13.906 ± 1.200	9.593 ± 2.857	N/A	6.041 ± 0.008	N/A	**4.161** ± 0.127	**3.722** ± 0.226	20.036 ± 0.705	18.402 ± 0.965	10.293 ± 1.595	**5.427** ± 0.114	**5.354** ± 0.389
**SE** **↑**	0.705	0.746	N/A	** 0.907 **	N/A	0.819	** 0.865 **	0.883	0.801	** 0.910 **	0.881	** 0.884 **
**SP** **↑**	0.972	0.973	N/A	0.982	N/A	** 0.983 **	** 0.991 **	0.969	0.981	0.980	** 0.986 **	** 0.987 **
**Sm** **↑**	0.844	0.890	N/A	0.869	N/A	** 0.912 **	** 0.915 **	0.807	0.892	0.885	** 0.916 **	** 0.919 **
**Em** **↑**	0.824	0.888	N/A	0.907	N/A	** 0.951 **	** 0.955 **	0.814	0.885	0.916	** 0.961 **	** 0.962 **
**MAE** **↓**	0.121	0.087	N/A	0.088	N/A	** 0.019 **	** 0.015 **	0.137	0.091	** 0.086 **	** 0.026 **	** 0.026 **

**Table 2 entropy-24-01755-t002:** Ablation analysis of our network on the FUSCC dataset. Dice similarity coefficient (DSC), Jaccard Index (JA), and Hausdorff distance (95%) (HD95) in test results are displayed in the form of mean ± standard deviation (SD) (95% confidence interval), and the best results are reported in **bold**. **(w/o W)**: with CM and without loss strategy for weighted pixels located at edges; *****: with CM module and loss strategy for weighted pixels located at edges. **↑:** larger is better, **↓**: smaller is better.

Index	Model	DSC↑	JA↑	HD95↓	SE↑	SP↑	Sm↑	Em↑	MAE↓
**(a)**	**backbone**	0.459 ± 0.011	0.298 ± 0.009	34.703 ± 3.417	0.558	0.889	0.739	0.719	0.178
**(b)**	**(a) +MF**	0.659 ± 0.010	0.491 ± 0.011	26.286 ± 2.303	0.720	0.907	0.825	0.876	0.101
**(c)**	**(b) +MD**	0.791 ± 0.003	0.654 ± 0.004	8.389 ± 1.228	0.854	0.958	0.875	0.908	0.076
**(d)**	**(c) +CM (w/o W)**	0.859 ± 0.001	0.753 ± 0.001	6.119 ± 0.388	0.864	0.978	0.910	0.952	0.034
**(e)**	**(c) +CM * (ours)**	**0.868** ± 0.001	**0.767** ± 0.002	**5.354** ± 0.389	**0.884**	**0.987**	**0.919**	**0.962**	**0.026**

**Table 3 entropy-24-01755-t003:** Performance comparison of different localization information fusions in LF (location fusion) on the FUSCC dataset. The best results are reported in **bold**.

Model	DSC	JA	HD95	SE	SP	Sm	Em	MAE
$f_{5}^{\prime }$	0.860 ± 0.002	0.754 ± 0.002	5.763 ± 0.341	0.864	**0.990**	0.917	0.956	0.027
$f_{4}^{\prime }+f_{5}^{\prime }$	**0.868** ± 0.001	**0.767** ± 0.002	**5.354** ± 0.389	**0.884**	0.987	**0.919**	**0.962**	**0.026**
$f_{3}^{\prime }+f_{4}^{\prime }+f_{5}^{\prime }$	0.866 ± 0.002	0.764 ± 0.002	5.219 ± 0.383	0.883	0.988	**0.919**	0.960	**0.026**

## Data Availability

Data sharing not applicable.

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
