# Peer review of "Improved Complementary Pulmonary Nodule Segmentation Model Based on Multi-Feature Fusion"

_entropy, 2022, doi:10.3390/e24121755_

Round 1
Reviewer 1 Report
1. The authors are suggested to elaborate more on the proposed components.
2. Similar explanations should be added in Section 5.1 and 5.2.
3. Is there some other public methods tested on the same datasets, if so, I'd like to see some comparisons with them.
4. The authors say: "We extracted pixels located at edges through an average pooling operation." in Section 3.4. Is it possible just using average pooling to do the edge detection, please present more details, and why not using canny or some other detector?
5. The figures' quality are required to improve, they are blurred when zoomed in.
6. Please move or replace some old and unrelated references.
Reviewer 2 Report
This paper presents a deep learning-based segmentation of lung nodules in CT images. The authors propose to use a coarse-to-fine segmentation strategy that combines high-level features with low-level features in order to exploit complementary information, especially edge information. A pre-trained Res2Net50 is used as the backbone of three-scale CT and 2D patches to capture coarse multi-scale features. The good results are obtained. To improve the paper, the authors should answer the following questions: 1) The loss function has two terms, one concerning the contour of the nodule and the other concerning its region. Different weighting coefficients, such as alpha and beta, and wi, are used to define the loss function. These weighting coefficients should be studied or discussed to show their impacts on the segmentation results. 2) How to choose the size of the CT patches? Does it influence the results? 3) Why did the authors not perform a 3D segmentation? Is it possible to extend the proposed method to 3D? 4) Why did the authors choose Res2Net50 as the backbone, and not the U-Net architecture as many other methods do?Author Response
Please see the attachment.

Round 2
Reviewer 1 Report
The authors answered my questions well.